# Vibration Prediction of the Robotic Arm Based on Elastic Joint Dynamics Modeling

**DOI:** 10.3390/s22166170

**Published:** 2022-08-18

**Authors:** Jianlong Li, Dongxiao Wang, Xing Wu, Kai Xu, Xiaoqin Liu

**Affiliations:** 1Key Laboratory of Advanced Equipment Intelligent Manufacturing Technology of Yunnan Province, Kunming University of Science and Technology, Kunming 650500, China; 2School of Mechanical and Electrical Engineering, Kunming University of Science and Technology, Kunming 650500, China; 3Yunnan Vocational College of Mechanical and Electrical Technology, Kunming 650203, China

**Keywords:** flexible joint, prediction function, parameter discrimination, vibration prediction

## Abstract

The flexibility of the joint drive system of an industrial robot can cause vibration at the end part, which can lead to motion errors. A method to predict the vibration during the motion of the robot arm is proposed considering the robot joint flexibility. The method combines the internal transfer function of the drive system and the identification of parameters under external excitation. Firstly, the dynamics of the robot joint system are modeled by a double inertia elastic system. The joint system transfer function from the electromagnetic torque to the arm vibration is obtained according to the dynamics model. To solve the unknown parameters in the transfer function, a vibration dynamics model of the joint arm under the external forces on the arm is developed. According to this model, the equivalent stiffness, damping and load inertia of the joint can be obtained by the direct parametric method. Then, the vibration spectrum of the robot arm is derived from the motor electromagnetic torque and joint dynamics models were used to predict the vibration spectrum of the robot arm. The experiments were conducted on a single-joint robot testbed, and on an articulated industrial robot. In both experiments, the key parameters in the system were determined by impact experiments. Then, the vibration signal of the arm during the robot motion was obtained by electromagnetic torque prediction. The predicted vibration signals are analyzed in comparison with the actual vibration signals. The experimental results both show the validity of the vibration prediction.

## 1. Introduction

Each joint of an articulated industrial robot is a complex mechanical drive system. The stiffness of each part of the mechanism in this system is limited. Among them, the stiffness of the gearbox is the smallest in the transmission system [1,2]. The elastic factors in the system cause mechanical vibration, which in turn affects motion accuracy and accelerates the fatigue of the drivetrain. Monitoring robot vibration caused by joint flexibility has important value to improve the reliability of industrial robots [3,4].

The basic method of monitoring robot vibration is to install vibration sensors directly on the arm. This method is simple and convenient, but it interferes with the normal operation of the robot and is not conducive to the proper functioning of the robot. In addition, the method of monitoring vibration by installing sensors is costly. By modeling the dynamics of the robot joint servo system, the vibration generated by the joint flexibility can be predicted directly from the internal electrical parameters [5]. Yang et al. [6] modeled the motion of a robot with a flexible jointed steel arm and obtained an equation of motion to obtain the angle of the jointed arm using the motor drive torque as the input to the joint system. The arm vibration prediction was performed using this equation of motion, and the accuracy of the equation of motion was verified by simulation. Moberg et al. [7] developed an extended flexible joint model. The method considers not only the flexibility of the harmonic reducer but also the flexibility of the bearing and arm. The vibration of the flexible joint was analyzed. Zaher [8] modeled the dynamics of n rigid links and n flexible joints from the point of view of multi-body dynamics. The dynamical equations are solved numerically, and the residual vibrations are analyzed. These studies have demonstrated the feasibility of vibration prediction both theoretically and experimentally, but they need to be performed on a high-precision test bench. Actual industrial robots cannot meet this high precision and are affected by field noise. These factors will affect the accuracy of the vibration prediction of the actual industrial robot. It can be used for highly flexible robotic arms when modeling systems using aggregate parameter schemes, which will increase the applicability of these methods [9].

The simplification of the actual physical system is conducive to the establishment of the dynamic model of the system. Additionally, vibration prediction of the elastic systems relies on both the dynamic model and input excitation. The servo drive system can be equated to an inertia system [10]. System dynamics are more easily obtained using inertial systems. Yabuki et al. [11] equated the servo drive system to multi-inertia systems and then obtained the frequency response characteristics from motor torque to motor output speed. Sato [12] uses a two-inertia equivalent servo drive system to obtain the frequency response characteristics of the system from vibration torque to torsional torque. In [13], when the coupling shaft between the rotor and the load is stiff, the system is called a one-inertia system. The system inertia is identified by the adaptive algorithm based on the model reference.

Good results in joint vibration prediction depend on the accuracy of the joint parameters. Parameters such as stiffness and inertia are highly uncertain and need to be solved by identification methods. A lot of research has been carried out on the identification of external mechanical parameters in the equivalent model of the servo drive system [14,15]. Parameter identification methods can be divided into two categories: offline and online methods. Offline methods are based on the acquisition of system input–output data, processing of the data, and identification algorithms [16]. Online methods estimate the system parameters by measuring the data online during the robot’s motion [17]. Both methods require the recording of input–output data during the motion of the robot joint system. The parameters are then identified according to the corresponding identification method.

This paper focuses on the scope of offline technology and also aims to provide specific guidelines for accurate parameter identification methods for flexible articulated robots.

In the literature, a number of papers deal with the identification of robot joint parameters. Beineke et al. extract motor speed response data for different stages for a two-inertia system with nonlinear characteristics. Identify the friction and clearance of the non-linear part of the system obtained [18]. Östring [19] analyzed a direct parameter identification method modeled as a three-inertia system for a robotic joint system. The physical parameters (inertia, stiffness, damping, etc.), are identified directly using the MATLAB system identification toolbox and compared with the “black box” model structure identification method. For non-parametric identification, Pacas et al. proposed a method to obtain the resonant frequency of a two-inertia elastic system directly from frequency response tests. Bode diagram obtained by the Welch method using a special internal excitation signal excitation system [20,21]. Internal excitation methods allow better control of the excitation signal bandwidth. However, a special excitation signal must be fed through the control system. This is difficult to achieve in many commercial robotic systems. In [17], the parameters of the flexible articulated robot manipulator are estimated using a filtered regressor form. In order to evaluate the performance of the proposed parameter identification scheme, the least squares with forgetting factor parameter identification method was also performed. Kostić et al. employs a model parameter identification method based on a regression volume form of the system model to identify the parameters of a robot system with a tandem elastic drive [22]. Due to insufficient estimation of the friction model, the authors did not obtain accurate results. In [23], a regressor-based method is proposed for parameter identification. The method allows the estimation of all dynamic parameters, but only provides simulation results.

In this paper, a method combining the internal transfer function of the drive system and the parameter identification under external excitation is proposed for the vibration prediction of the articulated arm of industrial robots. This paper is organized into five sections. After a literature survey in Section 1 and Section 2 describes the dynamic modeling of the prediction function. Additionally, a model for the identification of joint parameters under external excitation is studied. Section 3 identifies the parameters of the single-joint robot test stand and performs vibration prediction. In Section 4, a brief discussion of the results obtained from industrial robots. The paper ends with some conclusions in Section 5.

## 2. The Dynamic Modeling of the Prediction Function

### 2.1. Vibration Modeling of the Articulated Robot Arm under Electromagnetic Torque

The structure of each joint servo system of an industrial robot is similar in that they all consist of three parts: the servo motor, the speed reducer, and the actuator, as shown in Figure 1. In the figure, $T_{e}$ is the electromagnetic torque, $\theta_{m}$ is the motor rotation angle, $T_{l}$ is the load torque, $\theta_{l}$ is the rotation angle of the arm.

The joint elasticity is modeled as a linear torsional spring according to Spong’s theory [24]. The stiffness of the torsion spring is the result of the equivalence of the stiffness of each part of the transmission system. The equivalent stiffness of the joint is calculated following Equation (1). The equivalent motor-side inertia and load-side inertia for the high-speed axis switching to the low-speed axis are calculated following Equations (2) and (3).
(1)$$K=\frac{1}{\frac{1}{K_{1}}+\frac{1}{K_{2}}+\cdots +\frac{1}{K_{i}}}$$
(2)$$J_{M}=\left(J_{m}+J_{s}\right)\cdot n^{2}$$
(3)$$J_{L}=J_{l}$$
where $K_{1}$,$K_{2}$, …, $K_{i}$ are the stiffness of each part, K is the equivalent stiffness of the joint, $J_{m}$ is the actual motor inertia, $J_{s}$ is the inertia of the speed reducer, $J_{l}$ is the load inertia (the load is the arm), $J_{M}$ is the equivalent inertia of the motor-side, $J_{L}$ is the load-side inertia, and n is the reduction ratio.

Typically, the joints of an industrial robot are modeled as a double inertia system, as shown in Figure 2. Based on the above analysis, the differential equation of the system is established as follows:
(4)$$\{\begin{array}{c} J_{M}{\ddot{\theta }}_{s}+b_{m}{\dot{\theta }}_{s}=nT_{e}-T_{S}\\ J_{L}{\ddot{\theta }}_{l}+b_{l}{\dot{\theta }}_{l}=T_{S}-T_{L}\\ T_{S}=K\left(\theta_{s}-\theta_{l}\right)+c\left({\dot{\theta }}_{s}-{\dot{\theta }}_{l}\right)\\ \omega_{m}={\dot{\theta }}_{m};\;\omega_{l}={\dot{\theta }}_{l};\;\omega_{s}={\dot{\theta }}_{s} \end{array}$$
where $T_{S}$ is the torque of the drive shaft system, $b_{m}$ and $b_{l}$ are the damping of the motor and the load, $\theta_{s}$ is the output angle of the speed reducer, c is the shaft system damping, α is the angle of the arm to the *y*-axis, L is the arm equivalent length, and G is the arm equivalent weight.

According to Equation (4), the control block diagram of the dual inertia elastic system can be obtained, as shown in Figure 3. In addition, the transfer function between the load speed and the electromagnetic torque of the motor can also be obtained, as shown in Equation (5).
(5)$$G_{s}\left(s\right)=\frac{\omega_{l}}{nT_{e}}=\frac{cs+K}{\left(J_{M}J_{L}s^{3}+\left(\begin{array}{c} J_{M}b_{l}+J_{L}b_{m}\\ +J_{M}c+J_{L}c \end{array}\right)s^{2}+\left(\begin{array}{c} J_{M}K+J_{L}K\\ +b_{m}c+b_{l}c+b_{m}b_{l} \end{array}\right)s+K(b_{m}+b_{l})\right)}$$

According to the differential property of the Laplace transform, as shown in Equations (6):(6)$$\{\begin{array}{c} L\left[\frac{d^{m}f\left(t\right)}{dt^{m}}\right]=s^{m}F\left(s\right)-\sum_{i=0}^{m-1}\Psi_{m-i-1}\\ \sum_{i=0}^{m-1}\Psi_{m-i-1}=\sum_{i=0}^{m-1}s^{m-i-1}f^{\left(m-i-1\right)}\left(0\right) \end{array}$$
where f(t) is the time domain function, F(s) is a function of the f(t) after Laplace transformation, and Ψ is the constant introduced in the numerator for the corresponding differential order.

When analyzing the dynamic characteristics of the system, the effect of constants is not considered and Ψ is taken to be equal to 0. The transfer function from the electromagnetic torque to the angular acceleration of the joint arm rotation can be obtained from the differential characteristic by Equation (6), as shown in Equation (7).
(7)$$G_{a}\left(s\right)=\frac{{\ddot{\theta }}_{l}}{nT_{e}}=\frac{cs^{2}+Ks}{\left(J_{M}J_{L}s^{3}+\left(\begin{array}{c} J_{M}b_{l}+J_{L}b_{m}\\ +J_{M}c+J_{L}c \end{array}\right)s^{2}+\left(\begin{array}{c} J_{M}K+J_{L}K+b_{m}c\\ +b_{l}c+b_{m}b_{l} \end{array}\right)s+K(b_{m}+b_{l})\right)}$$

The servo drive control system introduces motor speed feedback, and the closed loop speed makes the system have good stability. From Equation (7), it can be seen that the introduction of motor speed feedback introduces conjugate poles and conjugate zeros to the closed-loop system. The frequency of the conjugate pole is called the natural torsion frequency (NTF) $\omega_{NTF}$. The frequency of the conjugate zero point is called the anti-resonance frequency (ARF) $\omega_{ARF}$. If $b_{m}=0$, $b_{l}=0$, and c=0 are assumed, the $\omega_{NTF}$ and $\omega_{ARF}$ can be expressed as follows:(8)$$\omega_{ARF1}=\sqrt{\frac{K}{J_{L}}}$$
(9)$$\omega_{NTF1}=\sqrt{K\left(\frac{1}{J_{M}}+\frac{1}{J_{L}}\right)}$$

According to Equation (7), if the system parameters are determined and the electromagnetic torque of the motor is known, then the angular acceleration of the articulated arm can be obtained, as shown in Equation (10).
(10)$${\ddot{\theta }}_{l}=n\cdot T_{e}\cdot G_{a}\left(s\right)$$

The actual joint system prediction process needs to consider the effect of transmission efficiency. Therefore, the actual prediction function is obtained as shown in Equation (11).
(11)$${\ddot{\theta }}_{l}=\eta \cdot n\cdot T_{e}\cdot G_{a}\left(s\right)$$
where η is the transmission efficiency of the joint system.

In general, the electromagnetic torque can be calculated from the motor current and the encoder. If the system transfer characteristics are clear, the acceleration of the arm can be obtained. The parameters involved in the system transfer characteristic Equation (7) include motor measured inertia and damping, the load measured inertia and damping, the coupling stiffness of the drive shaft system and damping, and the gearbox reduction ratio. Among them, the load-side inertia and the drive shaft coupling stiffness are usually unknown and need to be identified by parameter identification methods.

### 2.2. Parameters Identification Model under the External Torque

The internal excitation method and the external excitation method are two common methods for parameter identification [25,26]. For the internal excitation method, specific signals must be fed to the control system and it is difficult to implement in many commercial robotic systems. The external excitation method consists of establishing the transfer function of the external excitation and vibration response and then implementing parameter identification by applying excitation. This method is not dependent on the control system. It is a common method in experimental modal analysis for structure. In this paper, the method of external excitation is introduced to determine the key parameters of the joint.

#### 2.2.1. The Single-Degree-of-Freedom Systems

When the motor shaft is fixed, the joint model in Figure 1 can be simplified to a single-degree-of-freedom oscillating system with only the joint arm rotating as shown in Figure 4. In the figure, $T_{L}$ is the external excitation torque.

In the static equilibrium condition:(12)$$K\varphi_{s}=GL\cos\alpha$$
where $\varphi_{s}$ is the spring initial angle of rotation.

Assume that the swing angle of the arm with external excitation $T_{L}$ applied is $\theta_{l}$. The result is obtained from the momentum moment theorem as follows:(13)$$J_{L}{\ddot{\theta }}_{l}+c\left({\dot{\theta }}_{l}-{\dot{\varphi }}_{s}\right)+K\left(\theta_{l}-\varphi_{s}\right)+GL\cos\left(\alpha +\theta_{l}\right)=-T_{L}$$
(14)$$J_{L}{\ddot{\theta }}_{l}+c{\dot{\theta }}_{l}+\left(K-GL\sin\alpha \right)\theta_{l}=-T_{L}$$

When the arm is in a tiny vibration state, the $\theta_{l}$ is small:(15)$$\cos\theta_{l}\approx 1$$
(16)$$\sin\theta_{l}\approx \theta_{l}$$

When combining Equations (15) and (16), the Laplace transformation of Equation (14) yields the transfer function of the arm vibration angle about the externally acting moment as follows:(17)$$G_{ts}\left(s\right)=\frac{\theta_{l}}{T_{L}}=\frac{-1}{J_{L}s^{2}+cs+K-GL\sin\alpha }$$

The natural vibration frequency is obtained according to Equation (17):(18)$$\varpi_{NTF2}=\sqrt{\frac{K-GL\sin\alpha }{J_{L}}}$$

Obviously, the natural vibration frequency of the joint system under the action of the applied torque is close to the value of the antiresonant frequency of the joint system under the action of the electromagnetic torque (see Equations (8) and (18)). The value of GLsinα is much smaller than the value of K, so the values of the two frequencies are approximately equal. When the effect of GLsinα on the system is neglected in the actual system, the system is the same as the single-degree-of-freedom system in Figure 5.

Using the differential nature (see Equation (6)), Equation (17) is differentiated twice to obtain the transfer function of the angular acceleration of the vibration with regard to the externally acting moment. Each differentiation process introduces a constant, $\Psi_{2}$ and $\Psi_{3}$, respectively. As follows:(19)$$G_{ta}\left(s\right)=\frac{{\ddot{\theta }}_{l}}{T_{L}}=\frac{-s^{2}+\Psi_{2}s+\Psi_{3}}{J_{L}s^{2}+cs+K-GL\sin\alpha }$$

The direct parameter method is to directly identify the physical parameters of the structure based on the system input and output data [27]. This method is simple and easy to implement, and there is no conversion error in the intermediate process. The physical motion parameters of the system can be obtained directly from the identification results. Good estimation performance can be obtained as long as the system itself is sufficiently excited. In the direct closed-loop identification setup, the form of the transfer function in Equation (20) can be obtained by the direct parameter method, as follows:(20)$$G_{i}\left(s\right)=\frac{b_{1}s^{2}+b_{2}s+b_{3}}{s^{2}+a_{1}s+a_{2}}$$
where
(21)$$\{\begin{array}{c} b_{1}=-\frac{1}{J_{L}}\\ a_{1}=\frac{c}{J_{L}}\\ a_{2}=\frac{K-GL\sin\alpha }{J_{L}} \end{array}$$

In order to obtain the mechanical parameters $J_{L}$, K and c of the system, it is necessary to substitute $a_{1}$, $a_{2}$ and $b_{1}$ into Equation (21). $a_{1}$, $a_{2}$ and $b_{1}$ are obtained.

By substituting the stiffness, inertia, and damping obtained from the determined parameters in the system and the above identification back to the joint arm vibration prediction function, an explicit expression for the transfer function model from the electromagnetic torque of the motor to the arm vibration can be obtained. Using the electromagnetic torque signal and this model, the arm vibration spectrum can be predicted.

#### 2.2.2. The Multi-Degree-of-Freedom Systems

The actual system is flexible in all parts. The greater the flexibility, the greater the contribution to the vibration of the system. When there is only one part of the system with large flexibility, the less flexible part can be neglected. In this case, the system can be considered a single-degree-of-freedom system. When there are multiple parts of the system that are relatively flexible, this system is considered to be a multi-degree-of-freedom system, as shown in Figure 5b. The above single-degree-of-freedom system (see Figure 4) can be simplified to the single-degree-of-freedom system shown in Figure 5a by ignoring the influence of the connecting rod.

Converts it into a combination of single-degree-of-freedom systems when dealing with multi-degree-of-freedom systems. In [25], a multi-degree-of-freedom system is decoupled into a combination of multiple single-degrees-of-freedom. Through the decoupling process, a complex multi-degree-of-freedom system is reduced to a combination of a set of single-degree-of-freedom systems. Neglecting the effect of GLsinα in the transfer function of the single-degree-of-freedom system (see Equation (19)), as follows:(22)$${\tilde{G}}_{ta}\left(s\right)=\frac{{\ddot{\theta }}_{l}}{T_{L}}=\frac{-s^{2}+{\tilde{\Psi }}_{2}s+{\tilde{\Psi }}_{3}}{\tilde{J}s^{2}+\tilde{c}s+\tilde{K}}$$

Multi-degree-of-freedom systems can be obtained using this combination of single-degree-of-freedom systems:(23)$$G_{TA}\left(s\right)={\tilde{G}}_{ta1}\left(s\right)+{\tilde{G}}_{ta1}\left(s\right)+\cdots =\frac{-s^{2}+{\tilde{\Psi }}_{21}s+{\tilde{\Psi }}_{31}}{{\tilde{J}}_{11}s^{2}+{\tilde{c}}_{11}s+{\tilde{K}}_{11}}+\frac{-s^{2}+{\tilde{\Psi }}_{22}s+{\tilde{\Psi }}_{32}}{{\tilde{J}}_{22}s^{2}+{\tilde{c}}_{22}s+{\tilde{K}}_{22}}\cdots$$

Equation (23) can likewise be written in the form of a combination of Equation (20), as follows:(24)$$G_{TA}\left(s\right)=G_{i1}\left(s\right)+G_{i2}\left(s\right)+\cdots =\frac{b_{11}s^{2}+b_{21}s+b_{31}}{s^{2}+a_{11}s+a_{21}}+\frac{b_{12}s^{2}+b_{22}s+b_{32}}{s^{2}+a_{12}s+a_{22}}\cdots$$

In the same way as solving the single-degree-of-freedom parameters in part Section 2.2.1, the parameters of each single-degree-of-freedom system after decoupling can be obtained according to Equations (23) and (24).
(25)$$\left[\begin{array}{ccc} b_{11} & b_{12} & \cdots \\ a_{11} & a_{12} & \cdots \\ a_{21} & a_{22} & \cdots \end{array}\right]=\left[\begin{array}{ccc} -1/{\tilde{J}}_{11} & -1/{\tilde{J}}_{22} & \cdots \\ {\tilde{c}}_{11}/{\tilde{J}}_{11} & {\tilde{c}}_{22}/{\tilde{J}}_{22} & \cdots \\ {\tilde{K}}_{11}/{\tilde{J}}_{11} & {\tilde{K}}_{22}/{\tilde{J}}_{22} & \cdots \end{array}\right]\Rightarrow \left[\begin{array}{ccc} {\tilde{J}}_{11} & {\tilde{J}}_{22} & \cdots \\ {\tilde{c}}_{11} & {\tilde{c}}_{22} & \cdots \\ {\tilde{K}}_{11} & {\tilde{K}}_{22} & \cdots \end{array}\right]=\left[\begin{array}{ccc} -1/b_{11} & -1/b_{12} & \cdots \\ -a_{11}/b_{11} & -a_{12}/b_{12} & \cdots \\ -a_{21}/a_{11} & -a_{22}/b_{12} & \cdots \end{array}\right]$$

For a multi-degree-of-freedom system, multiple sets of parameters can be identified. These parameters are the result of the decoupling of the matrix composed of the system parameters. After decoupling each single-degree-of-freedom system is independent. The parameters obtained by using each single-degree-of-freedom system and the motor parameters can be regarded as an independent two-inertia system. When vibration prediction is performed, the prediction function is the result of multiple two-inertia systems connected in parallel.

### 2.3. Summary

A method combining the internal transfer function of the drive system and the parameter identification under external excitation is proposed for the vibration prediction of the articulated arm of industrial robots. Figure 6 shows the flow chart of the vibration prediction process, which can be summarized as follows:

(1)The dynamics model of the servo system is established under the action of external forces when the motor is stationary and holding the brake.(2)Using the shock signal to excite the system of stationary holding brake to obtain the system frequency response function (FRF).(3)The unknown parameters of the system are determined by the direct parameter method.(4)The transfer relationship from the electromagnetic torque of the motor to the vibration acceleration of the load when the motor is in motion is determined. Additionally, the vibration prediction function is established according to this transfer relationship.(5)When the servo system moves, the electromagnetic torque signal of the motor is obtained.(6)The vibration prediction function is determined based on the identification parameters and known parameters.(7)The spectrum of predicted vibration is determined by the motor’s electromagnetic torque signal and the vibration prediction function.

## 3. Experiments on a Single-Joint Test Bench

### 3.1. Experimental Setup

The single-degree-of-freedom articulated arm robot was built to simulate the single joint motion condition of an industrial robot. The entire setup is shown in Figure 7. The robot includes a PLC controller, servo motor, RV reducer, base, and articulated arm. The structural parameters of the test bench are shown in Table 1.

### 3.2. Identify Stiffness and Load Inertia by Impact Test

In order to determine the key parameters in the joint, impact experiments were carried out. At the beginning of the experiment, the pendulum arm was adjusted to a position parallel to the ground. The impact excitation force was applied to the pendulum arm by the hammer excitation method, and the vibration acceleration signal of the pendulum arm and the force signal applied by the hammer were collected at the same time. The impact location and acceleration sensor position are shown in Figure 7, the experimental force hammer model is YDL-2 SN201206, vibration sensor PCB single axis 333B30 type acceleration sensor. The sampling frequency is 1024 Hz and the sampling time is 2 s. The vibration acceleration signal is converted to angular acceleration by multiple the radius from the shaft to the accelerometer as shown in Figure 8b. The torque signal is converted to the torque in the same manner, as shown in Figure 8a.

The torque generated by the external excitation force is used as the excitation signal of the joint system, while the arm vibration caused by the excitation force is used as the response signal of the joint system. The frequency response function of the system is obtained from the excitation signal and the response signal. In the experiment, the frequency response function is the result of averaging 20 experiments, as shown by the solid blue line in Figure 9.

The identification transfer model was obtained according to the experimental FRF in the 0–50 Hz range curve by the direct parametric method [27]. The transfer model is shown in Equation (26), and the transfer model curve is shown as the red dashed line in Figure 9.
(26)$$G_{ita}\left(s\right)=\frac{-0.331s^{2}-9501s+2.45\times 10^{4}}{s^{2}+19.79s+6.41\times 10^{4}}$$

The inertia, stiffness, and damping of this single-degree-of-freedom system are obtained from the set of Equations (21) and (26), as shown in Table 2. $\Psi_{2}$ and $\Psi_{3}$ in Equation (19) are the constants brought about by the differential characteristic (see Equation (6)). The specific values of $\Psi_{2}$ and $\Psi_{3}$ are related to the angle and time variation functions. These two values can be obtained in the identification of the physical model. Because the relationship between angle and time variation is unknown, the physical relationship between these two values and the system is unknown. These two values are not used in the identification process. The stiffness in the reference value is from the manufacturer. The inertia is an approximately calculated value. The damping is related to various factors such as transmission parts, working conditions, and lubrication. The reference damping in Table 2 is calculated according to the material damping formula [28].

### 3.3. Prediction Experiment of Swing Arm Vibration

According to Equations (7) and (11), the dynamics model for vibration prediction involves motor inertia, load inertia, the equivalent stiffness of the joint, and the shaft damping. The motor inertia is known, while the load inertia, the equivalent stiffness of the joint, and the shaft damping are identified in Section 2.2. The damping of both motor and load are very low in the actual system [29], but cannot be ignored completely in the prediction process. In this paper, both motor and load damping are chosen as 2 N·m·s/rad [30]. The parameters are brought into Equation (7) to obtain the transfer function between the electromagnetic torque and the acceleration of the joint arm, and the amplitude part of the frequency response are plotted in Figure 10.

#### 3.3.1. Vibration Prediction under Torque Control

In this experiment, the motor is running in torque control. The specific process was as follows: the joint arm was adjusted to a position with an angle of −60 degrees to the *X*-axis as the starting position. The motor was controlled with a sweeping torque signal of 4–80 Hz, and the movement time was 3 s. The electromagnetic torque was obtained by PLC software reading with an internal sampling rate of 500 Hz. The vibration acceleration signal during the arm motion movement is acquired by the acceleration sensor, and the sampling rate was 1024 Hz. Similarly, the signal of the linear acceleration is converted into angular acceleration, and the angular acceleration of the joint vibration is equal to the angular acceleration of the arm vibration. Figure 11a shows the electromagnetic torque of the motor from the PLC software, and Figure 11b shows the spectrum of this electromagnetic torque signal.

The transmission efficiency of this test stand from the motor to the articulated arm is about 70%. According to Equation (11), the transmission efficiency, electromagnetic torque spectrum, and frequency response characteristics are multiplied to obtain the arm acceleration vibration signal spectrum, as shown in the dashed line in Figure 12. The solid line in Figure 12 shows the spectrum of the measured vibration acceleration signal of the arm. It is obvious that the predicted arm vibration acceleration signal spectrum in the frequency range of 10–75 Hz is basically consistent with the measured vibration. The predicted values in the frequency range of 0–15 Hz are smaller than the experimental values, which may be caused by the error of low-frequency torque measurement during the experiment.

#### 3.3.2. Vibration Prediction under Speed Control

Experiment of the arm vibration prediction when the motor is under speed control, and the specific process was as follows: the pendulum arm was adjusted to the position of 0 degrees from the *X*-axis as the starting position. The motion was controlled by PLC to the position of 120 degrees from the *X*-axis and then returned to the starting position with a motion time of 4.8 s. The control of the motor was changed to speed control. The speed control command of the motor is shown in Figure 13, and the maximum speed is 246 rad/s.

The experimental procedure is the same as in Section 3.3.1. Figure 14a shows a full-cycle motor electromagnetic torque signal. The amplitude of this signal is between −1.9 and 2.4 N·m. Figure 14b shows the frequency spectrum of this torque signal.

The predicted vibration spectrum is obtained according to Equation (11) as shown in the red dashed line in Figure 15. The spectrum of the electromagnetic torque (see Figure 14b) has burrs in the whole frequency band. So, the prediction result obtained is not a smooth curve.

The solid blue line in Figure 15 shows the spectrum of the experimentally obtained vibration. The predicted spectrum is consistent with the experimental spectrum in the range of 0–60 Hz. The experimental spectrum is larger than the predicted spectrum above 60 Hz. On the one hand, the model does not take into account that higher-order intrinsic frequencies can lead to this situation. The closer the frequency band is to the higher-order intrinsic frequency, the greater the effect on the prediction results. Only the first-order intrinsic frequency is considered in the prediction process, resulting in the predicted value for this cross-section being smaller than the experimental value. On the other hand, the electromagnetic torque signal used in the prediction is read directly from the PLC software. However, the low sampling rate of the PLC software showed some distortion in the high-frequency section and therefore the predicted values for the high frequencies were smaller than the actual values.

## 4. Experiments on the Articulated Robot

### 4.1. Experimental Setup

The vibration prediction experiments achieved good results on a single joint test stand (see Figure 7). However, the actual industrial robot joint motion is not as simple as it could be. On the one hand, industrial robots require multiple joints to work together, each of which is an independent system. In robot work engineering, the joint systems interact with each other. On the other hand, the flexibility of the components in each individual joint is different, so the analysis of the joint needs to be simplified on a case-by-case basis. To achieve vibration prediction of the robot, experiments were conducted on the industrial robot in Figure 16 below. The structural composition of each joint of the robot is the same as the structure of the test bench in Figure 7, and the parameters of each part of the robot are shown in Table 3.

Two experiments were conducted on the second joint of an industrial robot. First, the impact experiments were carried out to determine the key parameters in the joint system. Then, the vibration prediction experiments were performed while the second joint was moving alone.

### 4.2. Experimental Setup

The impact experiments were performed on the robot in Figure 16. The experimental procedure was the same as in Section 2.2. During the experiment, the attitude of the robot was adjusted to the position shown in Figure 16. The experiment was conducted to determine the key parameters in the second joint system. The vibration signals and the applied force signals of the robot arm were collected. The vibration acceleration signals were converted to angular acceleration as shown in Figure 17b. The force signal was converted to a torque signal as shown in Figure 17a.

The impact moment applied to the joint arm is in the plane of the *x–y* coordinate system and the direction is perpendicular to the linkage. The frequency response of the system obtained from the hammering experiment is shown as a solid blue line in Figure 18. There are two distinct resonance peaks in the frequency response function.

The flexible links that exist in the joint systems of actual industrial robots are not unique. Virtually every component’s flexibility contributes to system vibration. Usually, the flexibility of a structure is inversely proportional to its stiffness. The more flexible structures respond to system vibrations in the lower frequency portion. The vibration generated when the stiffness of the component is high reacts to the high-frequency part. The vibration in the high-frequency part is not easily excited and is analyzed mainly for the low-frequency part in the analysis. The low frequency is more likely to be excited in real-world environments, causing the system to resonate. When there are multiple structures with better flexibility in the system, the frequency response function of the system will have multiple resonance peaks in the low-frequency segment.

The vibrations caused by joint flexibility are mainly concentrated in the low-frequency range, so the curves in the 0–50 Hz range in the FRF are chosen to identify the joint parameters. This impact experiment excites two resonant frequencies of the second joint. The case of these two resonance peaks is treated according to the theory in Section 2.2. A segmented frequency response function with two peaks can be considered a two-degree-of-freedom system. This two-degree-of-freedom system can be obtained by the combination of two single-degree-of-freedom systems. The FRF of the single-degree-of-freedom system corresponding to the two peaks was obtained by the direct parameter method, as shown in Figure 18. The results of the summation of the two FRFs are shown in the red curve in Figure 19.

The expression for the two single-degree-of-freedom systems is obtained as shown in Equation (27). The parameter values for each single-degree-of-freedom system can then be obtained, as shown in Table 4. These parameters correspond to the values after the decoupling transformation of the system parameters.
(27)$$G_{TA}\left(s\right)=G_{i1}\left(s\right)+G_{i2}\left(s\right)=\frac{-0.116s^{2}-59.97s-5.76\times 10^{6}}{s^{2}+7.11s+1.11\times 10^{4}}+\frac{-0.074s^{2}+2.461\times 10^{2}s-5.882\times 10^{4}}{s^{2}+6.062s+3.251\times 10^{4}}$$

### 4.3. Vibration Prediction of the Second Joint

In the vibration prediction experiment of the second joint of the robot, only the second joint drive command is given. The motion commands of the industrial robot are implemented by writing joint motion angle programs. The vibration signal generated by the joint and the three-phase current signal of the motor is also collected. The vibration signal acquisition equipment is the same as that used for the above experiments. The current signal is collected by the current sensor.

Torque in an industrial robot joint cannot be obtained directly from a torque sensor. Therefore, the electromagnetic torque is usually calculated from the current. In [31], assuming that the measured line current is $i_{a}\left(t\right)$, and its Hilbert Transform (HT) is denoted as $h\left[i_{a}\left(t\right)\right]$, then $i_{q}\left(t\right)$ and $T_{e}\left(t\right)$ can be obtained as follows:(28)$$i_{q}\left(t\right)=\left|h\left[i_{a}\left(t\right)\right]\right|=\left|\frac{1}{\pi }\int_{-\infty }^{\infty }\frac{i_{a}\left(\tau \right)}{t-\tau }d_{\tau }\right|$$
(29)$$T_{e}\left(t\right)=K_{t}i_{q}\left(t\right)$$
where $K_{t}$ is the torque constant, and $T_{e}\left(t\right)$ is the electromagnetic torque of the motor.

The electromagnetic torque is obtained by multiplying the current and the envelope of the motor torque constant [31]. Figure 20 shows the motor’s three-phase current signal. The electromagnetic torque signal is obtained according to Equations (28) and (29) as shown in Figure 21a. Figure 21b shows the spectrum corresponding to the electromagnetic torque.

The parameters of the joint system are obtained by separately identifying the two single-degree-of-freedom systems. The joint system is considered a result of the parallel connection of two two-inertia systems in the prediction process. The parameters obtained from the two single-degree-of-freedom systems and the motor parameters form the two-inertia system, respectively. The procedure for predicting the vibration spectrum of the arm during motion by the two-inertia system is the same as in Section 2.3. The second joint parameter used in the prediction process was determined in Section 3.2, as shown in Table 4. Figure 22a shows the amplitude-frequency characteristic curves of the two two-inertia systems connected in parallel. The results of the summation of the two curves are shown in Figure 22b.

The predicted spectrum obtained from the prediction function (Equation (11)) is compared with the experimentally obtained spectrum as shown in Figure 23. The frequency response function (see Figure 22b) is a smooth curve, but burrs in the spectrum of the electromagnetic torque signal (see Figure 21b) resulting in the noise in the predicted result.

In Figure 23, the predicted results are compared and analyzed with the experimental results. It is obvious that the overall trends of the two curves match well. In the 0–80 Hz band, the predicted value is slightly larger than the experimental value. The reason for this difference is that the second joint is also weakly influenced by the flexibility of the other joints when operating alone.

## 5. Conclusions

In this study, the vibration problem of the arm caused by joint flexibility in industrial robots is addressed. First, the dynamics of the servo-flexible joint system are modeled. Then, a vibration prediction function using the electromagnetic torque of the motor and a dual inertia model is developed. Finally, the prediction function is used to predict the vibration of the arm. In addition, the uncertainty of the system parameters is addressed. A joint parameter identification model under the action of the applied torque is established. The parameters are identified by the direct parameter method in the case of external excitation. Experiments show that the proposed method of predicting the robot arm vibration by the prediction function is feasible. The frequency spectrum of the joint arm vibration signal is obtained using the motor electromagnetic torque signal. Additionally, the prediction accuracy is high in the frequency band close to the intrinsic frequency. The resonant frequency and vibration amplitude of the system are monitored by predicting this frequency band of the signal. This has practical implications for improving the life of industrial robots and monitoring the health of joints.

## Figures and Tables

**Figure 1 sensors-22-06170-f001:**
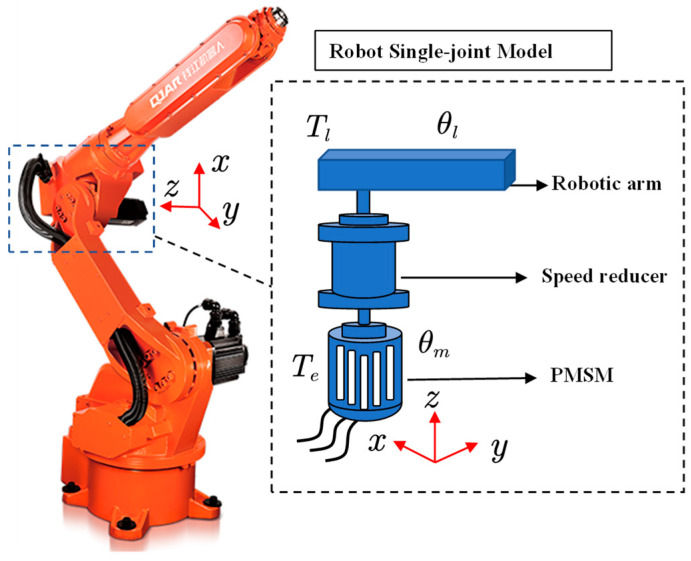
The schematic diagram of a single joint of an industrial robot.

**Figure 2 sensors-22-06170-f002:**
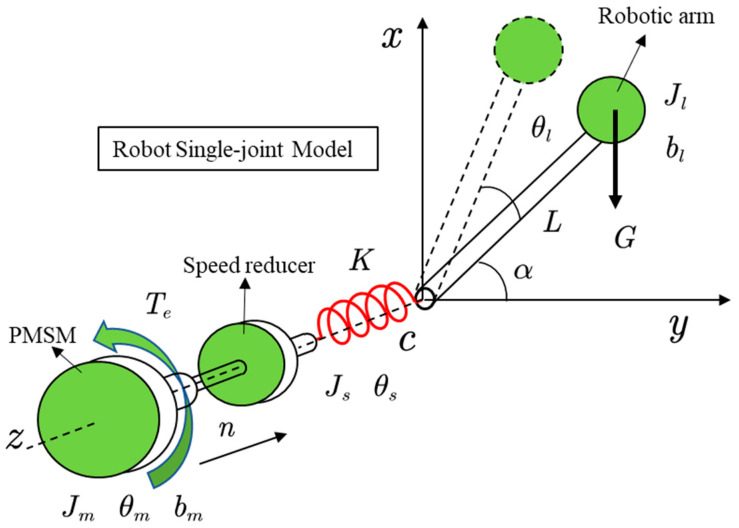
The double inertia model of the robot joint.

**Figure 3 sensors-22-06170-f003:**
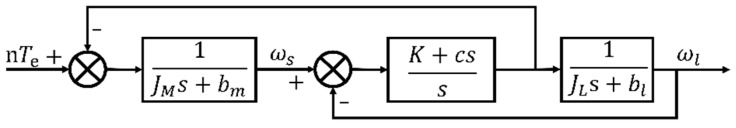
Control block diagram of the two-inertia elastic system.

**Figure 4 sensors-22-06170-f004:**
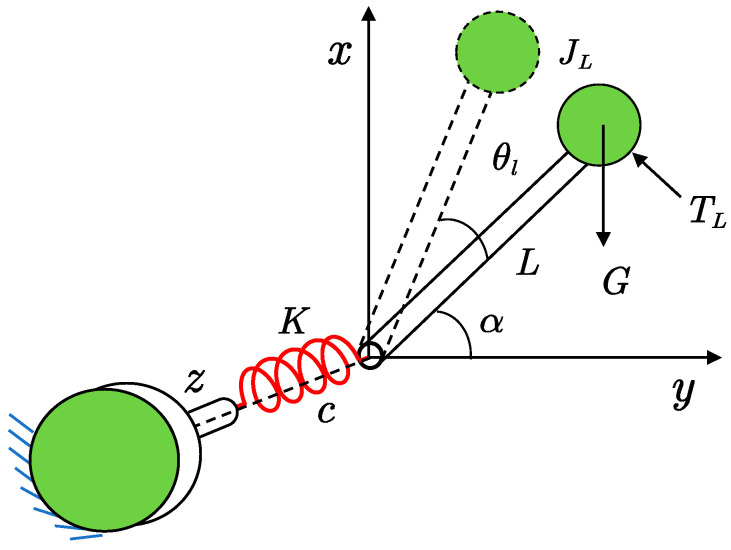
The single-degree-of-freedom oscillation system.

**Figure 5 sensors-22-06170-f005:**
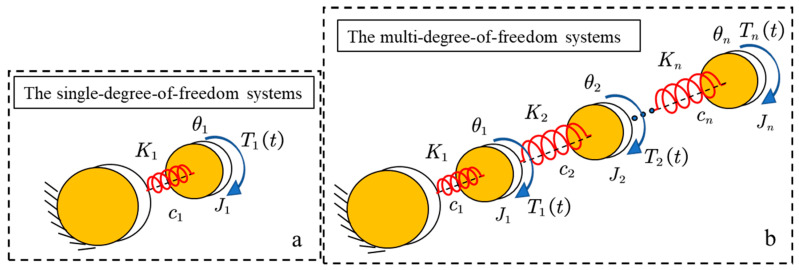
Simplification Model. (**a**) the single-degree-of-freedom; (**b**) the multi-degree-of-freedom.

**Figure 6 sensors-22-06170-f006:**
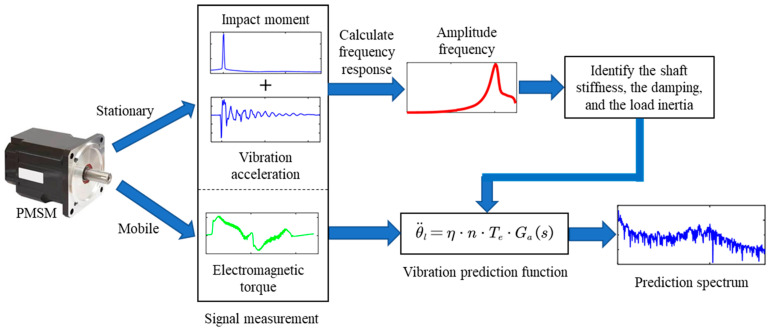
The flow chart of the prediction method.

**Figure 7 sensors-22-06170-f007:**
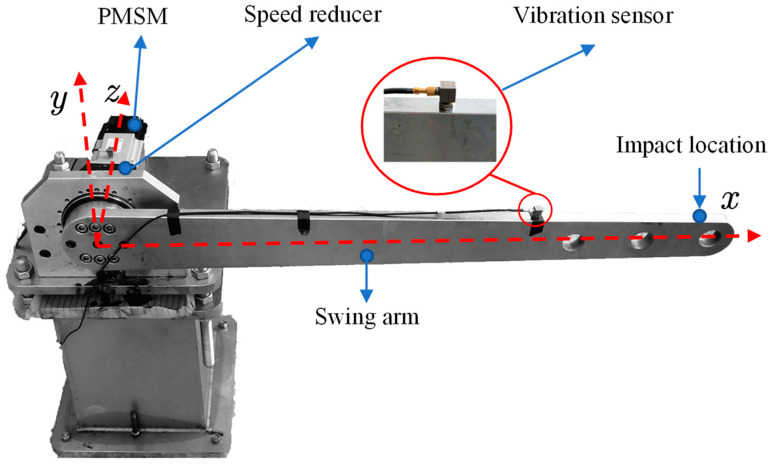
The single-degree-of-freedom rotary arm test stand.

**Figure 8 sensors-22-06170-f008:**
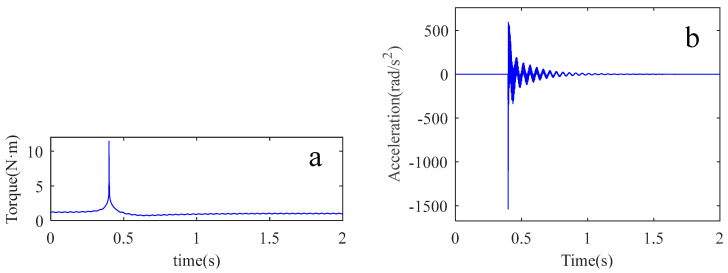
The results of the impact experiment. (**a**) impact torque signal; (**b**) vibration acceleration signal.

**Figure 9 sensors-22-06170-f009:**
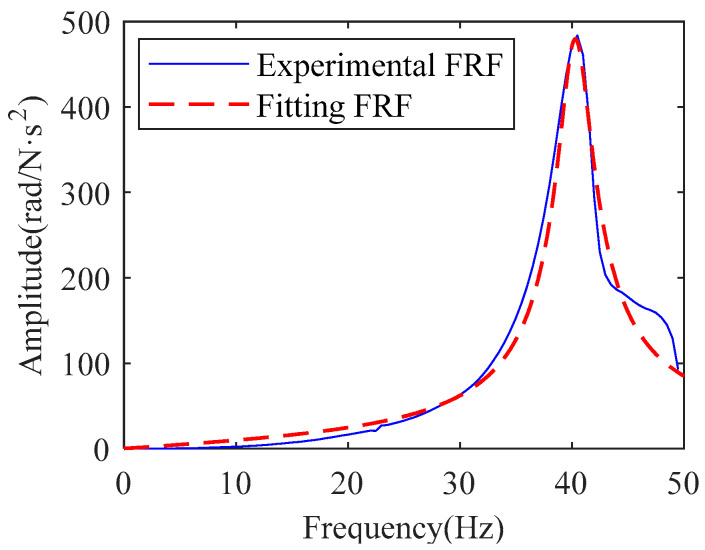
Frequency response function of articulated arm under hammering excitation.

**Figure 10 sensors-22-06170-f010:**
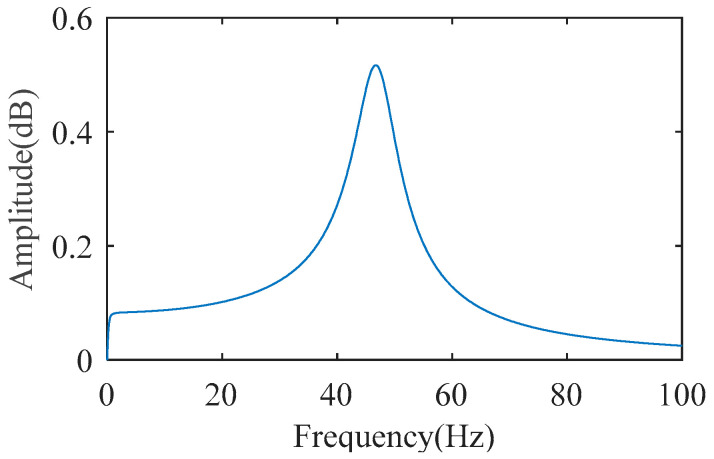
Amplitude–frequency characteristics of the transfer function from electromagnetic torque to arm vibration.

**Figure 11 sensors-22-06170-f011:**
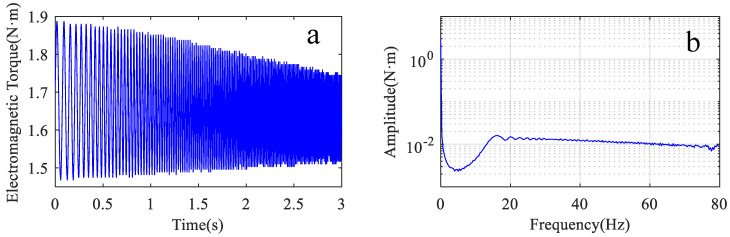
Motor electromagnetic torque waveform and frequency spectrum. (**a**) waveform of the signal; (**b**) spectrum of the signal.

**Figure 12 sensors-22-06170-f012:**
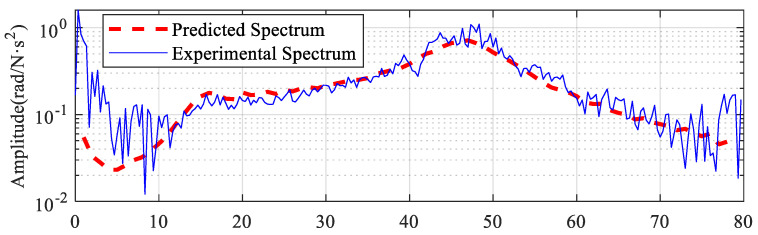
Comparison of predicted spectrum and actual spectrum.

**Figure 13 sensors-22-06170-f013:**
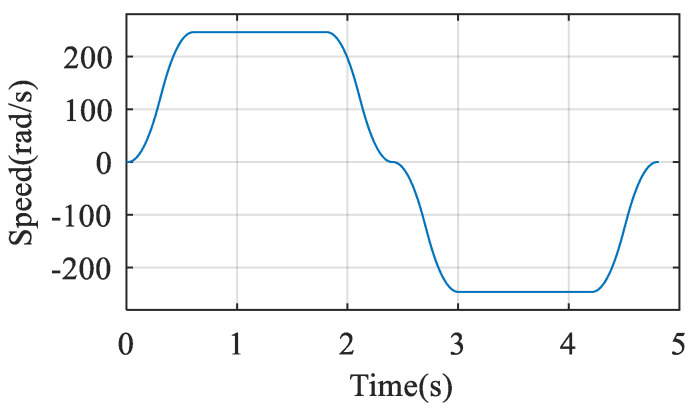
The speed control command of the motor.

**Figure 14 sensors-22-06170-f014:**
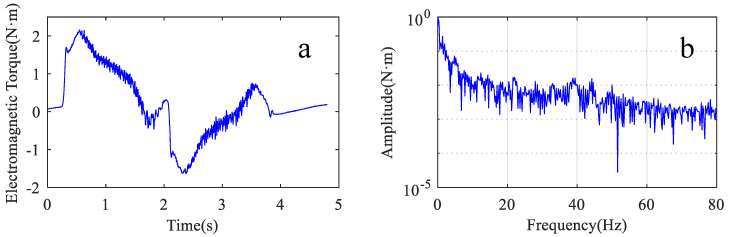
Motor electromagnetic torque signal. (**a**) waveform of the signal; (**b**) spectrum of the signal.

**Figure 15 sensors-22-06170-f015:**
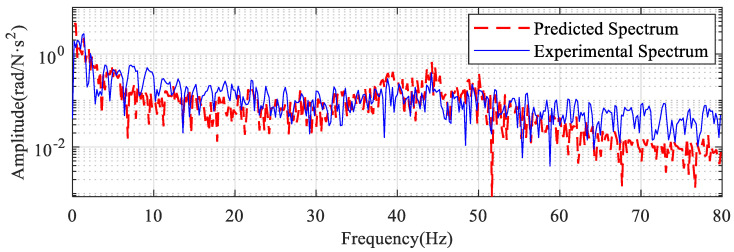
Comparison of predicted spectrum and experimental spectrum.

**Figure 16 sensors-22-06170-f016:**
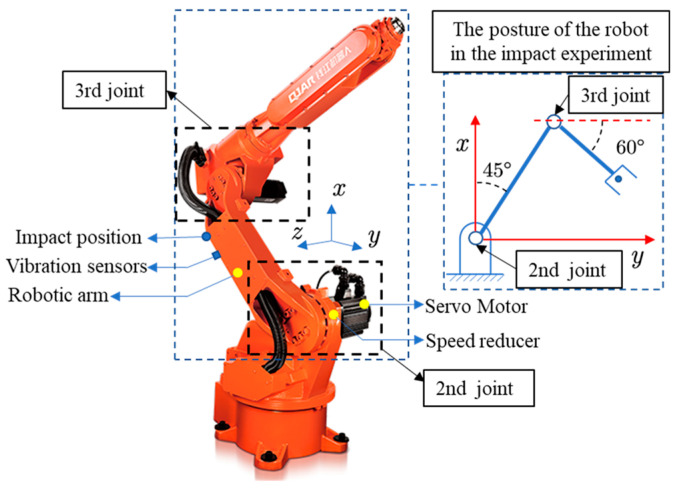
Six-degree-of-freedom industrial robots.

**Figure 17 sensors-22-06170-f017:**
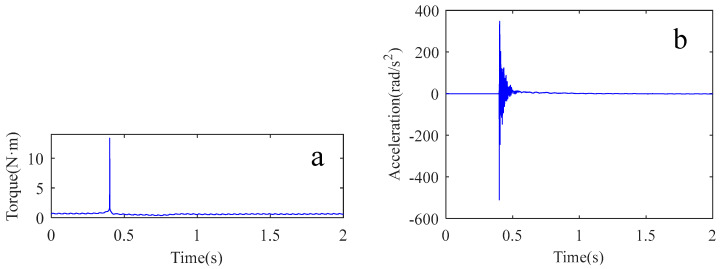
The results of the impact experiment. (**a**) impact torque signal; (**b**) vibration acceleration signal.

**Figure 18 sensors-22-06170-f018:**
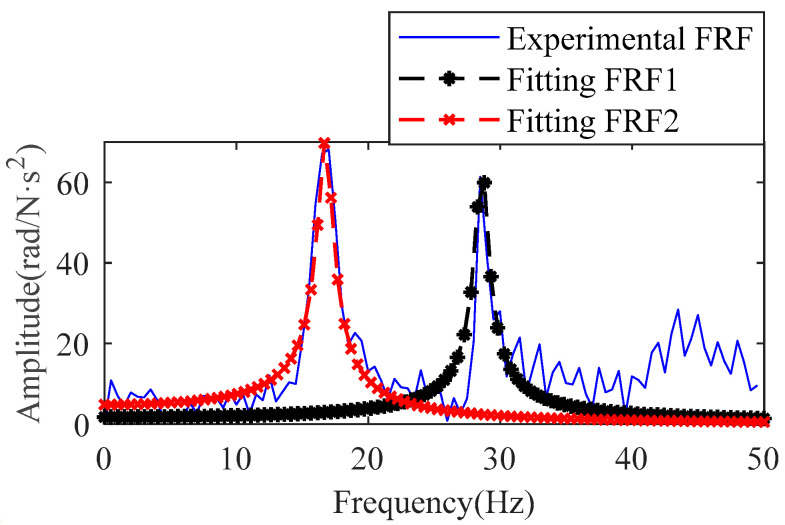
Two single-degree-of-freedom systems.

**Figure 19 sensors-22-06170-f019:**
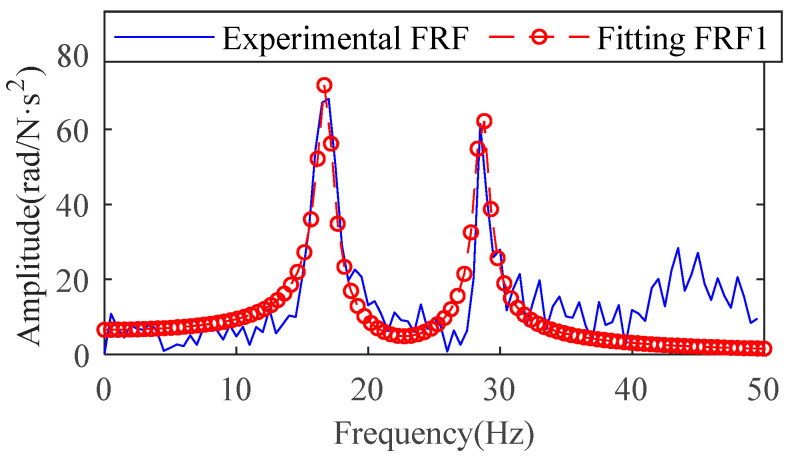
The result of adding two single-degree-of-freedom systems.

**Figure 20 sensors-22-06170-f020:**
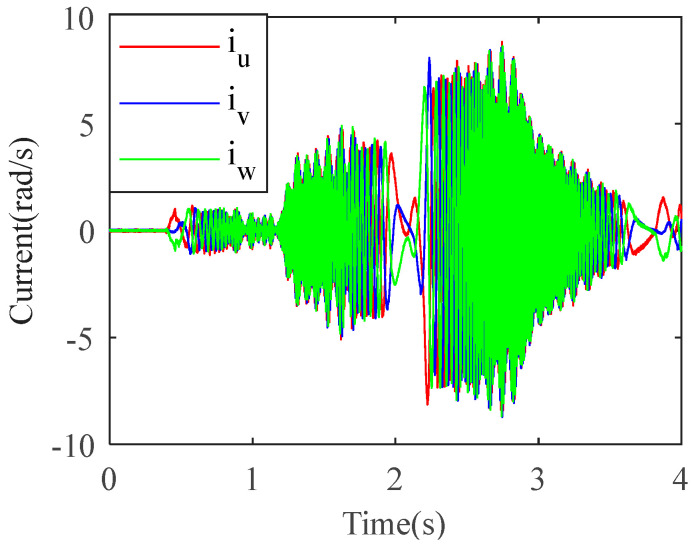
Three-phase current of the motor.

**Figure 21 sensors-22-06170-f021:**
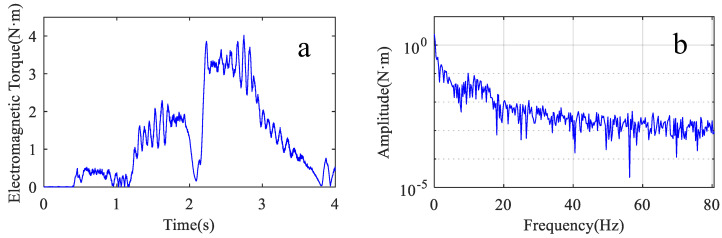
Motor electromagnetic torque. (**a**) waveform of the signal; (**b**) spectrum of the signal.

**Figure 22 sensors-22-06170-f022:**
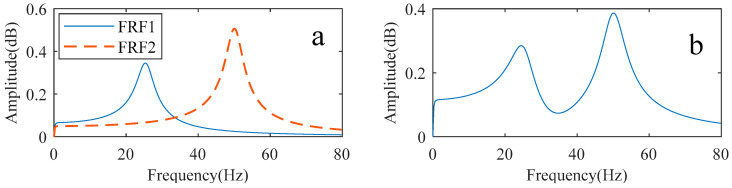
System amplitude and frequency characteristics. (**a**) Two dual inertia systems; (**b**) The results of the parallel connection of two dual inertia systems.

**Figure 23 sensors-22-06170-f023:**
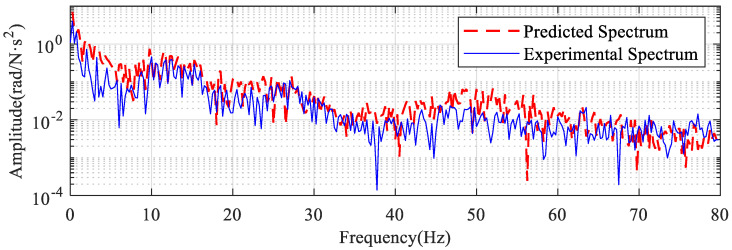
Comparison of predicted spectrum and actual spectrum.

**Table 1 sensors-22-06170-t001:** Test bench parameters.

Parameter	Value	Parameter	Value
PMSM rated power	1 kW	Swing arm length	1 m
Rotational inertia of PMSM Rotational inertia of reducer	0.00021 $\mathrm{kg}\cdot m^{2}$ 0.000248 $\mathrm{kg}\cdot m^{2}$	Reducer stiffness Swing arm mass	168,449.59 N⋅m/rad 12 kg

**Table 2 sensors-22-06170-t002:** Identification results.

Parameters	Identifying Value	Reference Value
Stiffness	193,665 N⋅m/rad	168,449 N⋅m/rad
Inertia	3.02 $\mathrm{kg}\cdot m^{2}$	3.4 $\mathrm{kg}\cdot m^{2}$
Damping	59.76 N⋅m⋅s/rad	30.27 N⋅m⋅s/rad

**Table 3 sensors-22-06170-t003:** Test bench parameters.

The Parameters of the Second Joint	Value
PMSM rated power	2 kW
Rotational inertia of PMSM	0.001 $\mathrm{kg}\cdot m^{2}$
Rotational inertia of reducer	0.00032 $\mathrm{kg}\cdot m^{2}$
Number of motor pole pairs	5
Torque constant of the motor	0.458

**Table 4 sensors-22-06170-t004:** Identification results.

**Parameter**	Identifying Value	Identifying Value
Stiffness	${\tilde{K}}_{11}=95689.66\;N\cdot m/\mathrm{rad}$	${\tilde{K}}_{22}=439324.32\;N\cdot m/\mathrm{rad}$
Inertia Damping	${\tilde{J}}_{11}=8.603\;\mathrm{kg}\cdot m^{2}$ ${\tilde{c}}_{11}=61.29\;N\cdot m\cdot s/\mathrm{rad}$	${\tilde{J}}_{22}=13.55\;\mathrm{kg}\cdot m^{2}$ ${\tilde{c}}_{22}=81.92\;N\cdot m\cdot s/\mathrm{rad}$

## Data Availability

Not applicable.

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
