# Peer review of "Vibration Prediction of the Robotic Arm Based on Elastic Joint Dynamics Modeling"

_sensors, 2022, doi:10.3390/s22166170_

Round 1
Reviewer 1 Report
In this article, the author proposes a method for predicting the vibration of the robot arm during the movement process considering the flexibility of the robot joints. It establishes the internal transfer function of the drive system and the parameter identification under external excitation. And the validity of the vibration prediction is verified on the single-joint robot test bench and the joint industrial robot. In order to improve the paper, I have the following comments:
(1)In this article, lines 113, 179, and 281 do not match the format of the rest of the similar paragraphs in the article.
(2)In this article, table 4 on line 465 has a different format than the previous table.
(3)In the fourth part of the experiment on the joint robot, the uncertainty of the interaction of the joint system is not considered. The experiment only predicts the vibration of the second joint, and does not consider the influence of the third joint on it. I hope the author can continue Complete.
(4)As shown on line 249, the paragraph 1.2.1 described in the following paragraph should be revised to paragraph 2.2.1 to meet the meaning of the article.
(5)The description of Figure 8(a) in the paragraph is a force signal, while the following caption describes an impact torque signal, and the two are inconsistent.
(6)In the experimental part, the selected actual industrial robot has a small number of joints (only three joints), and it is recommended to increase the vibration prediction experiment of industrial robots with a higher number of joints.
(7)The text below Figure 2 is separated from Figure 2 and should be adjusted.
(8)The letter b_m and b_i in line 118 does not appear in formula (4) and should be corrected.
(9)The format of the text below the full text picture is confusing and should be unified.
(10)The letter w in lines 145 and 146 are inconsistent with the formats in formulas (8) and (9), and should be unified.
(11)The title of Section 2.2 is repeated with the title of Section 2.1 and the content of Section 2.2 does not match the title, please correct it.
(12)In the single-joint experiment, the effect of higher-order intrinsic frequencies appears when the frequency is greater than 60Hz, but this effect does not appear in the experiments on the articulated robot, please explain.

Author Response
We thank the reviewers for suggesting changes to our paper. We have followed the list of comments to revise the manuscript. Please see the attachment. Below is an example of one of the revisions.
Point 5: The description of Figure 8(a) in the paragraph is a force signal, while the following caption describes an impact torque signal, and the two are inconsistent.
Response 5: Thanks again for your comments. The signal in the figure and the signal in the description part are all impact torque signals. We re-modify the contents of the description of Figure 8(a). As follows, we illustrate the changes before and after the modification.
The original manuscript: The force signal is converted to the torque in the same manner, as shown in Figure 8a.
The revised manuscript: The torque signal is converted to the torque in the same manner, as shown in Figure 8a.

Reviewer 2 Report
The authors propose a method to estimate the vibrations induced by the motion in an industrial robot. The system is considered as a multi-body rigid arm with flexibility concentrated in the joints. They use the knowledge of the transfer function of the drive system of a joint in combination with a direct parameters identification using external excitation. Firstly, they analyze the case of a two-degrees of freedom to illustrate the method. Then, they use the same strategy to analyze a multi-degrees of freedom system. Finally, the case of an industrial robot is investigated to prove the feasibility of the method. Comparisons with numerical predictions and experimental data are provided for validating purposes.
The reviewer finds no fault whatsoever with the methods, analysis, discussions, and conclusions. It is an interesting contribution, and hence it could be published but after some minor revisions. See the points below.
The English must be improved a bit, in some cases, the style is too choppy. Please read carefully the text and amend it where needed. In many sentences, there are the same words repeated too much. It is preferable to avoid starting a sentence with the conjunction "And". this is logically inconsistent! "And" cannot follow a "full stop" since it is the link between two parts of a sentence with the same importance/role.
On page 2 line 62, "obtains" must be "obtained" because in the first part of the sentence the past tense is used.
On page 2 line 61, please specify what means "multi-inertia systematic", here there are only adjectives without a noun, so it is unclear what it means.
On page 2 line 78, Please correct "Mario et al.", it should be "Pacas et al.", Mario is the first name, not the surname.
Please correct in Table 1 the unit of measure of power is not "KW" but must be "kW", it is kilowatt, not kelvin watt! The same mistake is found in table 3.
On page 11 line 350, please correct "motion movement" the two words have the same meaning, so they are repeated twice.
Please explain the meaning of HT on page 16 line 475. Please never use an abbreviation without explanation. The same goes for FRF. Please, the first time that you use it, define it.
The introduction is too short, a more detailed literature review must be provided especially about the parameter identification (see for instance [1]).
[1] Miranda-Colorado, R., & Moreno-Valenzuela, J. (2018). Experimental parameter identification of flexible joint robot manipulators. Robotica, 36(3), 313-332.
The method is very interesting and the reviewer suggests mentioning also the fact that it can be used for highly flexible robot arms when they are modeled with lumped parameter scheme (see for example [2]). This will increase the applicability of the method.
[2] Giorgio, I. & Del Vescovo, D. (2019) Energy-based trajectory tracking and vibration control for multi-link highly flexible manipulators. Mathematics and Mechanics of Complex Systems 7(2): 159–174.
Author Response
We thank the reviewers for suggesting changes to our paper. We have followed the list of comments to revise the manuscript. Please see the attachment. Below is an example of one of the revisions.
Point 4: On page 2 line 78, Please correct "Mario et al.", it should be "Pacas et al.", Mario is the first name, not the surname.
Response 4: We re-modify the contents of line 78. As follows, we illustrate the changes before and after the modification.
The original manuscript: Mario et al……
The revised manuscript: Pacas et al……
